# Parasitic nematode fatty acid- and retinol-binding proteins compromise host immunity by interfering with host lipid signaling pathways

**Sophia C. Parks**[1], **Susan Nguyen**[1], **Shyon Nasrolahi**[1¤], **Chaitra Bhat**[1], **Damian Juncaj**[1], **Dihong Lu**[1], **Raghavendran Ramaswamy**[2], **Harpal Dhillon**[1], **Hideji Fujiwara**[3], **Anna Buchman**[4], **Omar S. Akbari**[4], **Naoki Yamanaka**[5], **Martin J. Boulanger**[2], **Adler R. Dillman**[1]*

1 Department of Nematology, University of California, Riverside, California, United States of America,
2 Department of Biochemistry and Microbiology, University of Victoria, Victoria, British Columbia, Canada,
3 Division of Endocrinology, Metabolism, and Lipid Research, Washington University School of Medicine, St. Louis, Missouri, United States of America, 4 Division of Biological Sciences, Section of Cell and Developmental Biology, University of California, San Diego, California, United States of America,
5 Department of Entomology, University of California, Riverside, California, United States of America

¤ Current address: Keck School of Medicine, University of Southern California, Los Angeles, California, United States of America
* adlerd@ucr.edu

**Data Availability Statement:** All relevant data are within the manuscript and its supporting files. All raw data are available on Mendeley Data; Dillman,

## Abstract

Parasitic nematodes cause significant morbidity and mortality globally. Excretory/secretory products (ESPs) such as fatty acid- and retinol- binding proteins (FARs) are hypothesized to suppress host immunity during nematode infection, yet little is known about their interactions with host tissues. Leveraging the insect parasitic nematode, *Steinernema carpocapsae*, we describe here the first *in vivo* study demonstrating that FARs modulate animal immunity, causing an increase in susceptibility to bacterial co-infection. Moreover, we show that FARs dampen key components of the fly immune response including the phenoloxidase cascade and antimicrobial peptide (AMP) production. Our data also reveal that FARs deplete lipid signaling precursors *in vivo* as well as bind to these fatty acids *in vitro*, suggesting that FARs elicit their immunomodulatory effects by altering the availability of lipid signaling molecules necessary for an efficient immune response. Collectively, these data support a complex role for FARs in immunosuppression in animals and provide detailed mechanistic insight into parasitism in phylum Nematoda.

## Author summary

A central aspect of parasitic nematode success is their ability to modify host biology, including evasion and/or subversion of host immunity. Modulation of host biology and the pathology caused by parasitic nematodes is largely effected through the release of proteins and small molecules. There are hundreds of proteins released by nematodes during

Adler (2021), "Parks et al. 2020 FAR proteins", Mendeley Data, V1, doi: 10.17632/7ccth7hz72.1 (https://data.mendeley.com/datasets/7ccth7hz72/1).

**Funding:** This research was supported by a USDA National Institute of Food and Agriculture Hatch project (accession #1015192) and R35 GM137934 National Institute of General Medical Sciences to A. R.D. and by a Canadian Institutes of Health Research Grants 148596 to M.J.B. M.J.B. gratefully acknowledges the Canada Research Chair program for salary support. O.S.A and A.B. were supported in part by an NIH Award # DP2AI152071. The funders had no role in study design, data collection and analysis, decision to publish, or preparation of the manuscript. The funders had no role in study design, data collection and analysis, decision to publish, or preparation of the manuscript.

**Competing interests:** The authors have declared that no competing interests exist.

an infection and few have been studied in detail. Fatty acid- and retinol-binding proteins (FARs) are a unique protein family, found only in nematodes and some bacteria, and are released during nematode infection. We report that nematode FARs from *S. carpocapsae*, *C. elegans* and *A. ceylanicum* dampen fly immunity decreasing resistance to infection. Mechanistically, this is achieved through modulation of the phenoloxidase cascade and antimicrobial peptide production. Furthermore, FARs alter the availability of lipid immune signaling precursors *in vivo* and show binding specificity *in vitro*.

## Introduction

Helminths cause significant morbidity and mortality on a global scale through human disease, sickening of livestock, and a reduction in crop yield [1,2]. Parasitic nematodes are responsible for significant human suffering with a striking 1.5 billion people infected with soil-transmitted helminths alone [3]. Much of the nematodes' success in parasitism is due to their ability to disguise themselves from host immune defenses. Despite the importance of these organisms to human and animal health, little is understood about the molecular mechanisms that underpin these stealth processes. Increasing our understanding of immune modulation by parasitic nematodes has the significant potential to inform treatment of such infections and lead to therapeutics for immune dysregulation such as autoimmune disorders.

Protein and small-molecule effectors are crucial for parasites to successfully infect plants, invertebrates, and vertebrate hosts [4,5]. Upon infecting their hosts, nematodes release excretory/secretory products (ESPs) into surrounding tissues, which are the primary point of interaction between the parasite and host and are hypothesized to aid in nematode survival and cause damage to the host. A variety of proteins are found in nematode ESPs, some of which are known to cause tissue damage and modulate host immunity. One family of proteins released by parasitic nematodes is the fatty acid- and retinol-binding protein family (FARs). FARs are small (~20 kDa), alpha-helix rich, and may be responsible for sequestering essential lipids required for reproduction and development [6]. FARs were first discovered in the filarial nematode *Onchocerca volvulus*, the causative agent of river blindness. FARs have subsequently been identified in the free-living nematode *Caenorhabditis elegans* as well as several other nematode parasites of animals, plants and insects [7–9]. In *C. elegans*, FARs are localized in the hypodermis, head and lips which are regularly in contact with the outside environment, as well as the excretory cell connected to the hypodermis [10].

FARs have attracted attention because they were shown to be secreted by parasitic nematodes and are strongly immunogenic in infected hosts [8,11–13]. Moreover, several studies suggested that FAR proteins modulate plant immunity. The strongest evidence comes from a study on a FAR protein from the plant-parasitic nematode *Meloidogyne javanica*, which, when expressed *in planta*, has been shown to increase the host susceptibility to further infection [14]. FARs are released by infectious-stage parasites, and therefore several studies have suggested the importance of FARs in facilitating parasitism by sequestering host retinol and/or fatty acids interfering with or suppressing immunity [15,16]. Previous research papers on FAR proteins from animal-parasitic nematodes suggest that FARs modulate animal immunity, yet no studies have investigated this experimentally. The functional role of these proteins in animal hosts as well as the mechanism of action behind these functions remains unknown.

One of the challenges to studying these host-parasite interactions is the paucity of studies in model systems to develop testable hypotheses about effector function during infection. Using an insect-parasitic nematode model allows studies to be conducted on large populations and

some insect-parasitic nematode species are closely related to vertebrate-parasitic species, leading to insights that may be directly applicable to mammalian diseases [17,18]. Entomopathogenic nematodes (EPNs) are parasites of insects that are related to skin-penetrating nematodes of mammals such as *Strongyloides stercoralis* [19–21]. Using the EPN *Steinernema carpocapsae* as a model parasite and the fruit fly *Drosophila melanogaster* as a model host provides a robust system to characterize potential effectors such as FARs. The fruit fly is a powerful genetic model with a complex innate immune system containing mechanisms conserved in mammals [22]. The *D. melanogaster* immune response has been a powerful model for mammalian immunity and is divided into two innate response types; cellular and humoral immunity [23,24]. Furthermore, immune responses are generally activated by two main NF-kB signaling pathways, Toll and Imd, similar to human toll-like receptors (TLR) and tumor necrosis factor (TNF) signaling respectively [23]. The Toll and Imd pathways are activated by different pathogens, depending on properties such as cell wall constituents. The cellular response is carried out by hemocytes and involves processes that surround and kill the invading pathogen including phagocytosis and encapsulation, whereas the humoral response involves the generation of antimicrobial peptides (AMPs) by the fat body, analogous to the mammalian liver and adipose tissues. AMPs are small, cationic and often amphipathic peptides that function as endogenous antimicrobials that "mop up" pathogens left over from the cellular immune defense system by disrupting negatively charged microbial membranes. Systemic production of AMPs is controlled by the two NF-kB pathways, Toll and Imd [24]. A combinatorial response by the innate immune system is likely elicited during a nematode infection [25].

In this study, we tested the hypothesis that nematode FAR proteins (Sc-FARs) can dampen the animal immune response (*D. melanogaster*) in the context of various bacterial infections. This work is the first to demonstrate that nematode FARs are detrimental to the outcome of infection and are directly immunomodulatory in an animal system. Significant immune suppression by FARs was observed resulting in decreased host survival to bacterial infections, accompanied by a significant increase in microbe growth. To understand how Sc-FARs modulate host immunity several readouts of *Drosophila* immunity were assessed including AMP production and phenoloxidase activity. Potential FAR binding partners were assessed using metabolomics and *in vitro* protein-metabolite interactions. In addition to demonstrating the potent immunomodulatory effect of FARs in an animal, these data led us to propose a model for nematode FAR modulation of host immunity.

## Results

### Genomic analysis of S. carpocapsae revealed an expansion of FAR proteins

Analysis of the genome sequence of the generalist insect-parasitic nematode *S. carpocapsae* revealed an expansion of FARs compared to other nematodes [20]. We evaluated the presence of FARs in a variety of parasitic nematodes and *C. elegans* and found a dynamic range of putative FAR-encoding genes among nematode genomes (Table 1). We found that *S. carpocapsae* had the most with 45 putative FAR proteins, compared to 9 in *C. elegans*, 16 in *S. stercoralis* and 5 in *Ascaris lumbricoides* (Table 1). While FARs are thought to be essential for lipid sequestration, we found no putative FAR-encoding genes in either *Trichinella spiralis* or *Trichiuris muris*, suggesting that either these nematodes have divergent putative FAR genes that were not recognizable by sequence similarity, or that they have evolved a different strategy to acquire lipids from their hosts. Of the 45 putative FARs in *S. carpocapsae*, 5 FARs were found in the ESPs of an *in vitro* infection model [26]. We chose the 2 found in highest abundance, Sc-FAR-1 (L596_023208) and Sc-FAR-2 (L596_016036), for further study.

**Table 1. Number of putative FAR-encoding genes present in the genomes of various nematode species.** Species in order from left to right: *Steinernema carpocapsae* (Scar), *Ancylostoma ceylanicum* (Acey), *Strongyloides stercoralis* (Sste), *Ascaris lumbricoides* (Alum), *Brugia malayi* (Bmal), *Caenorhabditis elegans* (Cele), *Dracunculus medinensis* (Dmed), *Haemonchus contortus* (Hcon), *Onchocera volvulus* (Ovol), *Nippostrongylus brasiliensis* (Nbra), *Trichinella spiralis* (Tspi), *Trichiuris muris* (Tmur). *T spiralis* and *T. muris* have no known FAR encoding genes while *S. carpocapsae* shows the greatest expansion of FAR genes in its genome.

| | *FAR encoding genes* | | | | | | | | | | | |
|---|---|---|---|---|---|---|---|---|---|---|---|---|
| *Spp.* | *Scar* | *Acey* | *Sste* | *Alum* | *Bmal* | *Cele* | *Dmed* | *Hcon* | *Ovol* | *Nbra* | *Tspi* | *Tmur* |
| **# of genes** | 45 | 24 | 16 | 5 | 3 | 9 | 3 | 11 | 3 | 12 | 0 | 0 |

### Nematode FAR proteins modulate host immunity decreasing resistance to bacterial pathogens

To investigate the immunomodulatory effects of FARs, we used *D. melanogaster* as a model host and the two most abundant ESP-derived *S. carpocapsae* FARs Sc-FAR-1 and Sc-FAR-2, hereafter referred to as FAR-1 and FAR-2, respectively. We tested *D. melanogaster*'s susceptibility to *S. carpocapsae* nematode infection but found that when flies are infected with any number of infective juveniles IJs, they die quickly, and none are able to recover (S1 Fig). This makes is difficult to observe immune modulation during an *S. carpocapsae* infection in flies. Therefore, to measure immune modulation by FARs in flies, we utilized a bacterial infection and first determined the $LD_{30}$ dose of the Gram-positive, extracellular pathogen, *Streptococcus pneumoniae* in OregonR flies and established the appropriate dose for injection and the baseline outcome of infections at this dose (S2 Fig). The outcome of infection was assessed using fly survival and microbial growth over time. We found that the $LD_{30}$ (the dose that kills 30% of adult flies in the first 2–7 days post infection) is the optimal dose for injection, since it provides sufficient sensitivity to detect shifts in the outcome of infection. Recombinant FAR-1 and FAR-2 were co-injected into adult flies, mimicking an EPN infection with delivery of the protein and pathogenic bacteria into the hemocoel. These experiments revealed that a one-time dose of FAR-1 and/or FAR-2 co-injected with the $LD_{30}$ dose of *S. pneumoniae* significantly affects the outcome of bacterial infection in flies. A dose-dependent effect was observed where a high dose (250 ng) decreased host survival while lower doses did not elicit the same effect (Fig 1A and 1B). We found no additive effects of FAR-1 and FAR-2 after co-injection (S3 Fig).

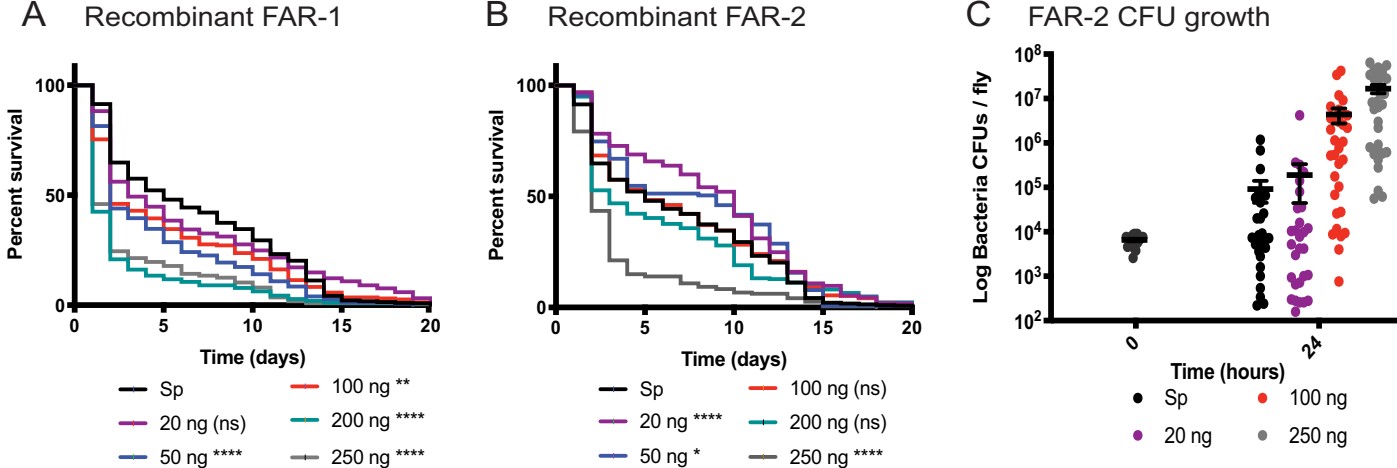

**Fig 1. Recombinant FARs reduced fly survival in a dose-dependent manner.** A&B) 7,000 CFUs of *S. pneumonia* (*Sp*) were co-injected with various nanogram doses of either Sc-FAR-1 or -2. C) CFUs were measured at 24 hours post-injection for panel B which shows a trend of increased bacterial growth with increasing protein doses. All controls for survival curves (black) are *Sp*-injected only, without the addition of FAR proteins. Time 0 CFUs representative of all fly strains. Log-rank test p-value significance indicated by asterisks on Kaplan Meier graphs. CFU graphs show p-value significance of an unpaired t-test (error bars show mean+SEM). Survival curves n≥180, CFU graph n≥24. All raw data available in supplemental materials.

We measured microbe growth over time for FAR-2 recombinant injections due to its dose-dependent spread of effects on the outcome of survival. A trend in increased microbial load was also observed, where the flies injected with the highest dose of FAR had a greater number of bacterial cells 24-hours post-injection (Fig 1C). Heat-denatured recombinant FAR-1 or FAR-2 was co-injected at the highest dose tested (250 ng); no difference was observed when compared to *S. pneumoniae*-only injected flies, confirming that the addition of folded, recombinant FAR in a one-time dose alters the flies' ability to deal with pathogenic infection (S4 Fig).

In addition to a one-time injection of recombinant of FAR-1 and FAR-2, we also determined the outcome of infection using transgenic flies expressing either FAR-1 or FAR-2. We ectopically expressed these FARs in flies using the Gal4/UAS system and confirmed expression with Western blot (S5 and S6 Figs). We also evaluated the effects of *C. elegans*-derived FAR-8 and a FAR from the human hookworm *A. ceylanicum* (S7 Fig). Flies expressing nematode FAR proteins in the fat body, hemocytes, and salivary glands in the absence of bacterial infection are healthy and have a normal lifespan (S8 Fig). An expression-dependent decrease in survival in a bacterial infection model, was observed with strong, ubiquitous expression resulting in the most severe decrease in survival. Expression of FARs only in the hemocytes had the least severe effect on the outcome of infection (Fig 2A). The effect observed on survival is specific to FAR-expressing flies since their genetic control did not elicit the same effect (S9 Fig). In addition, overexpression of mCherry, a protein irrelevant to the system, does not have any effect on outcome of bacterial infection (S10 Fig). FAR-expressing flies also displayed a significant increase in microbial load 24 hours post-injection (Fig 2B). To determine whether this effect was pathogen-specific, the Gram-negative EPN symbiont *Xenorhabdus innexi*, one of the least virulent bacteria in the genus, was also injected to induce an immune response [27]. Similar to *S. pneumoniae* infection alone, flies expressing FAR-1 had a significant decrease in survival for both doses, whereas FAR-2-expressing flies were only affected when injected with the higher dose (Fig 2C). CFUs were also measured post-injection. A significant increase in microbial load was observed in both FAR-1 & 2-expressing flies (Fig 2D). The intracellular Gram-positive pathogen *Listeria monocytogenes* was also tested. We found FAR expressing flies to have the same decrease in survival, however there was no significant difference in microbe load 24 hours post-injection (Fig 2E and 2F). These data demonstrate that FARs significantly affect the outcome of a bacterial infection, in a dose dependent manner, by decreasing survival post infection and dampening the ability of the flies to control the bacterial growth over time. Interestingly, flies expressing *C. elegans* FAR-8 and a FAR from the human hookworm *A. ceylanicum* showed a similar immune deficient phenotype when challenged with an *S. pneumoniae* infection (S7 Fig). This suggests that the immunomodulatory effects of FARs are conserved across taxa, solidifying the value of this model for understanding the mechanism of FARs in parasitic infection.

## FARs dampen key aspects of immunity in D. melanogaster

*D. melanogaster* has a sophisticated, evolutionarily conserved, innate immune response that can be categorized into two main branches: a systemic humoral response, and a cellular response mainly carried out by various types of hemocytes (S11 Fig). To establish the mechanism by which FAR-1 and FAR-2 elicit their immunomodulatory effects, we evaluated several readouts of immunity including phenoloxidase (PO) activity and melanization, which are essential immune defenses during a bacterial infection. Disseminated melanization and PO activity was measured post-injection with *L. monocytogenes*, which elicits a robust disseminated melanization phenotype 4 days post injection [28]. Flies expressing FAR-1 and FAR-2

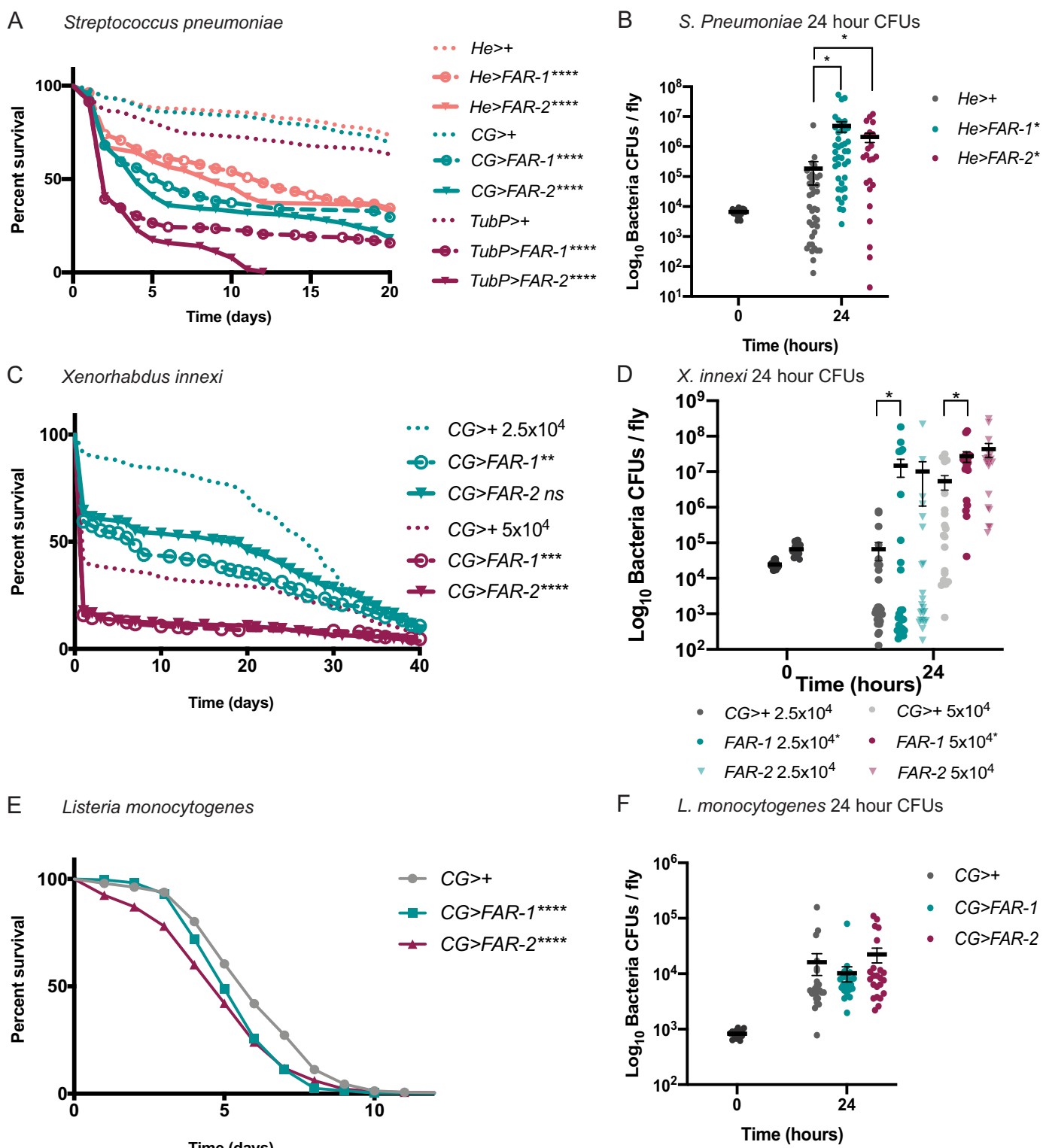

**Fig 2. Transgenic FAR expression significantly decreases fly survival and increases bacterial load 24-hours post-injection.** Flies were injected with specified doses of various pathogens. Survival was monitored daily and bacterial load was measured 24 hours post-injection. A&B) Flies FAR-1 & -2 expressed with a tubulin driver (TubP), a fat body, hemocyte and lymph gland specific driver (CG), and a hemocyte only driver (He) were injected with 7,000 CFUs of *Streptococcus pneumoniae*. Survival curves of flies are shown in (A) and bacterial CFUs in the He driver flies at 24-hour post injection in (B). C&D) Flies with FAR-1 & -2 expression with the CG

driver were injected with 25,000 and 50,000 CFUs (shown at time 0) of *Xenorhabdus innexi*. Survival curves of flies are shown in (C) and bacterial CFUs in (D). E&F) Flies with FAR-1 & -2 expression with the CG driver were injected with 1,000 CFUs of *Listeria monocytogenes*. Survival curves of flies are shown in (E) and bacterial CFUs in (F). All controls are 86Fa flies crossed with flies containing the specified driver. Time 0 CFUs representative of all fly strains (depicted in black). Statistics shown as log-rank test p-value for survival curves and unpaired t-test for microbe growth with (error bars show mean+SEM). Survival curves n≥180, CFU graph n≥24 over at least 3 experiments. All raw data available in supplemental materials.

showed a significant decrease in survival when injected with ~1,000 CFUs of *L. monocytogenes* (Fig 2E). After injection with 1,000 cells of *L. monocytogenes*, FAR-1 and FAR-2 expressing flies were unable to initiate PO activity six hours post-infection (Fig 3A). Similarly, disseminated melanization, measured as two or more melanization spots throughout the body, was reduced to only 20%, from 60% in controls, of the population in FAR-expressing flies 4 days post-injection (Fig 3B). To further investigate the effect of FARs on innate immunity we assessed their effect on the production of AMPs. We performed RT-qPCR to measure expression levels of two AMP-encoding genes: *Defensin*, mostly regulated by the Imd pathway, and *Drosomycin*, mostly regulated by the Toll pathway [29]. After infection with *S. pneumonia*, both FAR-1 and -2 expressing flies exhibit at least a 14-fold reduction in the AMP response for both *Drosomycin* and *Defensin* (Fig 3C). Collectively, these data demonstrate that FARs dampen key aspects of immunity.

## FARs alter in vivo fatty acid availability including putative immune signaling molecules

To determine the molecular mechanism underlying the immunomodulatory effects of FAR-1 and FAR-2, we initially screened for potential metabolites available to FAR using an

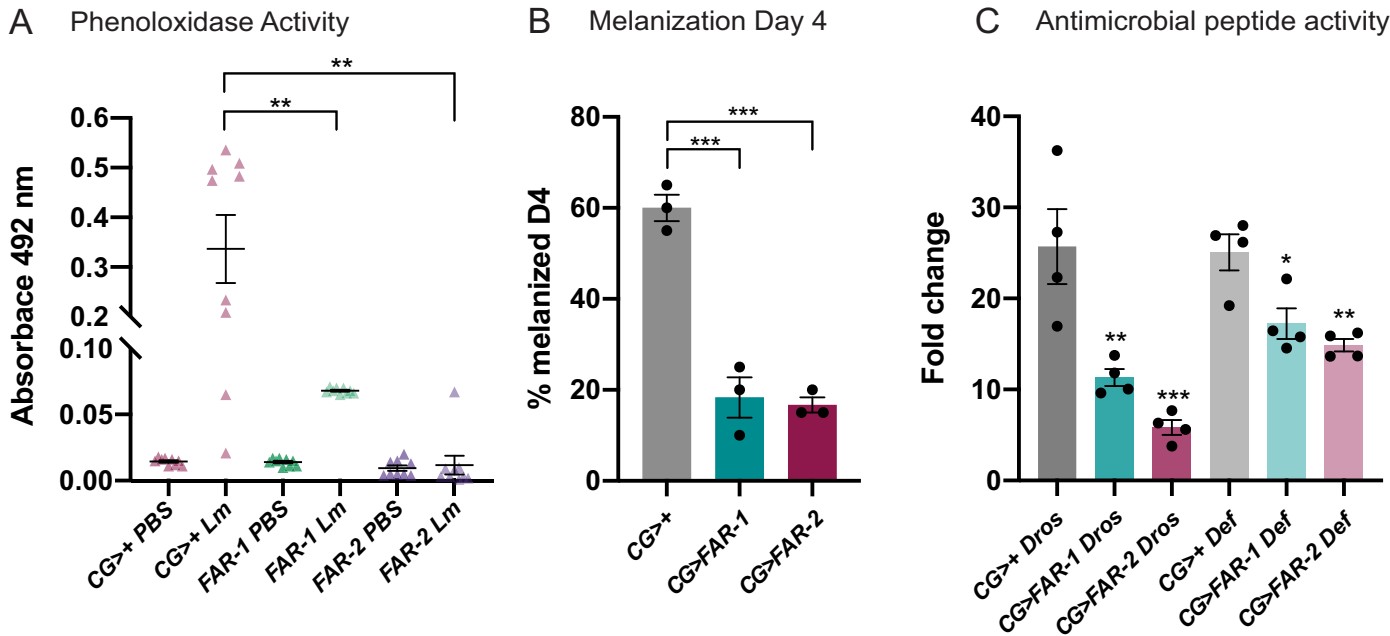

**Fig 3. Melanization, phenoloxidase and antimicrobial peptide activity was diminished in FAR expressing flies.** A) Flies were injected with the $LD_{30}$ of *L. monocytogenes* (1,000 CFUs) and phenoloxidase activity was measured 6 hours post-injection. B) Flies were injected with the $LD_{30}$ of *L. monocytogenes* and disseminated melanization was observed 4 days post-injection. C) 24 hours post-injection with the $LD_{30}$ dose of *S. pneumoniae* the relative increase in the antimicrobial peptides (AMP) *Drosomycin* (Toll) and *Defensin* (Imd) were analyzed. Compared to the CG>+ control, FAR-1 and -2 significantly suppress both AMPs. Four experiments were completed with 15 flies per experimental group. Asterisks indicate statistical significance from one-way ANOVA (A & C) and unpaired t-test (B). All experiments were repeated at least 3 times, error bars show mean+SEM. All raw data available in supplemental materials.

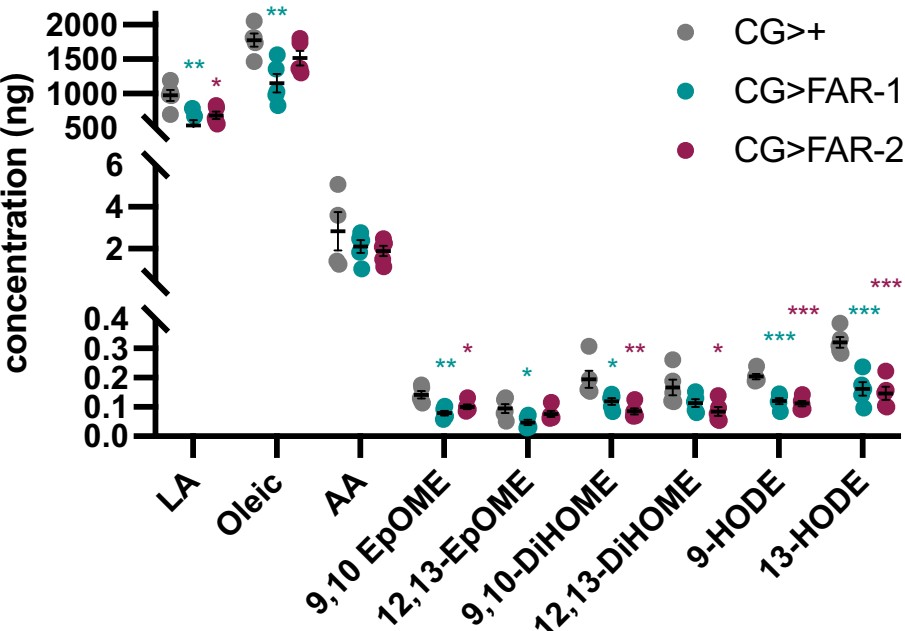

**Fig 4. Metabolite abundance in FAR expressing and control hemolymph.** A) Fly hemolymph samples were analyzed for abundance of various lipids. 9(10)- and 12(13)- EpOME, 9- and 13-HODE and arachidonic acid were depleted in FAR-1 expressing flies. Statistics shown as an unpaired t-test and error bars show mean+SEM. Asterisk color for significance matches sample color. All raw data available in supplemental materials.

untargeted metabolomics approach. We found many phosphatidylcholines (PC), phosphatidylethanolamines (PE), as well as multiple fatty acids that were less abundant in FAR-expressing flies, suggesting they are somehow depleted by FARs (S12 Fig). Since many of the molecules that were significantly altered were unclassified, we moved to a targeted metabolomics approach. During an infection, FARs are released into the host tissue and may remain in circulation to elicit their immunomodulatory effects. Therefore, we proceeded to test differences in lipid metabolite abundance in the hemolymph of the FAR-1- and FAR-2-expressing flies, as well as control flies. Hemolymph was collected from 200 male flies and lipid concentrations between groups were assessed with targeted metabolomics. We found that several fatty acid metabolites, some of which are epoxide derivatives of linoleic acid and leukotoxins in mammals, were significantly lower in FAR-expressing flies when compared to the genetic control (Fig 4). These included linoleic acid, 9,10-EpOME, 9,10-DiHOME, 9-HODE, and 13-HODE. In addition to considerable overlap in lipid depletion, FAR-1 and FAR-2 affected some lipids differently. For example, oleic acid and 12,13-EpOME were depleted in FAR-1-expressing flies, but not FAR-2-expressing flies.

## FARs have measurable differences in binding specificity in vitro

To determine potential binding partners of FARs, we measured the binding affinity to various fatty acids and retinol *in vitro*. Initially, we determined the binding affinity of both Sc FARs to the 11-(Dansylamino) undecanoic acid (DAUDA), and retinol, by calculating the equilibrium dissociation constant (Kd) (S10A and S10B, and S11A and S11B Figs). We

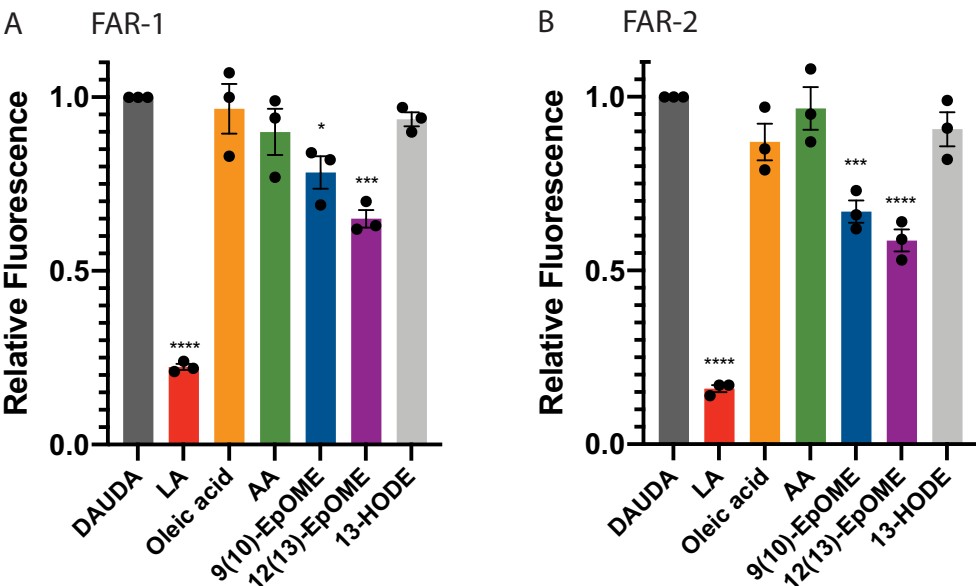

**Fig 5. In vitro binding of Sc-FARs.** A&B) Competitive binding between 10-fold excess linoleic (LA), oleic, arachidonic (AA) acid, and other fatty acids (all 10 μM) and DAUDA (1 μM) was tested in the context of FAR-1 & 2. Linoleic acid displaces DAUDA the most, as seen by the largest reduction in relative fluorescence. Both 9(10)- and 12(13)-EpOME displace DAUDA more easily in the presence of FAR-2, showing tighter binding. Statistics shown as One-way ANOVA and error bars show mean+SEM. All raw data available in supplemental materials.

found the Kd to be in the 1–10 μM range (S13C and S13D, and S14C and S14D Figs). Using DAUDA as the fluorophore, we tested various fatty acids in 10-fold excess in competition with DAUDA to determine preferred fatty acid binding partners for FARs *in vitro*. When FAR binds to a fatty acid in this assay, DAUDA is displaced from the protein resulting in a reduction in peak fluorescence intensity (S13E and S13F Fig). Linoleic acid caused the greatest displacement of DAUDA and 9(10)- and 12(13)-EpOME disrupted DAUDA bound to FAR-2 greater than FAR-1 (Fig 5). The results indicated that FAR-1 & 2 have measurable differences in binding specificity, suggesting that they may target different lipid signaling molecules during infection.

## Discussion

The results of this study provide the first evidence demonstrating that FARs alter the animal immune response. Understanding how parasitic nematodes circumvent host immune responses has significant potential to inform therapeutic intervention strategies. Much of what is known about FAR proteins and their potential interactions with host immunity comes from *in vitro* studies [30–34]. FARs initially attracted attention because they are excreted by parasitic nematodes during an active infection and elicit a host immune response [8,11–13]. A previous study has shown that FARs increase host susceptibility to nematode infections in plants however after performing *S. carpocapsae* nematode infections, we found that once a fly was infected with even one nematode it was unable to recover from the infection and died quickly [14]. This makes it especially difficult to determine variation in immune responses since the infection is rapidly lethal at any dose. To investigate FAR's effects on immunity we utilized bacterial infections and found that the outcome of infections are significantly worse in flies when FARs are present.

## FAR effects on innate immunity

We have investigated the specific aspects of immunity that are affected by FARs, and found modulation of various mechanisms including phenoloxidase activity, a key mechanism to control invading parasites, and antimicrobial peptide production. Upon injury or infection, insects rapidly initiate the melanization cascade leading to deposition of melanin along the wounding site and around invading microbes or pathogens. This serves to block invading pathogens, prevent excess hemolymph loss, and to encapsulate and kill pathogens with reactive oxygen species (ROS) and other toxins. Melanization is independent of the classical immune pathways (Toll & Imd). Rather, it is dependent upon activation of prophenoloxidase (proPO), which is a proenzyme that is cleaved by prophenoloxidase activating enzyme (PPAE) to its active form phenoloxidase (PO). Phenoloxidase is the catalyst for the oxidation of mono- and diphenols to orthoquinones, which form polymers with melanin non-enzymatically, during which ROS is produced [24,35]. Since the family of FAR proteins is highly conserved among many nematode species it is important to understand whether FARs elicit their immunomodulatory effects in similar ways (Table 1) [35]. Slight differences between the effects of the two Sc-FARs are apparent in flies challenged with *X. innexi*, where FAR-2 expressing flies given a 25,000 CFU dose did not do significantly worse than the control flies, as well as the recombinant studies and AMP production, pointing to a potential mechanistic specificity of FARs. We performed additive recombinant FAR-1 and FAR-2 co-injections and found no difference between a 250 ng dose of FAR-1 or FAR-2 and a 125 ng FAR-1 plus 125 ng FAR-2 combined dose showing that FAR does not have an additive effect. FAR-1 and 2 seem to have similar *in vivo* effects, at least at the level of the outcome of infection, leading us to believe that downstream immune effects are also being affected in similar ways not specific enough to evaluate their differences. We found that even a one-time dose of recombinant FAR has severe adverse effects on the outcomes of a bacterial infection. Flies generally exhibited a decrease in survival in a dose-dependent manner, beginning with a 50 ng dose of FAR-1. During a bacterial infection the host is harmed by two factors: the pathogen and the immune response. In experiments with FAR-2, low doses (20 ng and 50 ng) led to significantly improved survival, which we hypothesize is due to the suppression of immune-induced damage. Although 250 ng is likely greater than physiological concentrations of FAR during EPN infections, it is remarkable that such a phenotype was observed with only a single component of a complex array of venom proteins found in ESPs, where hundreds of proteins usually act in concert to dampen host immunity. Interestingly, FARs not only modulate measurable outputs of immunity but affect the outcome of infection.

## Interactions of FARs and host lipid signaling pathways

A key component of understanding how nematodes utilize FARs to dampen host immunity is to understand the mechanism underlying their interactions with immune molecules. We hypothesize that FARs bind to lipids that function as signaling molecules for a diverse range of functions including inflammation, immunity, homeostasis, and reproduction. It has been proposed that most terrestrial insects lack free long-chain polyunsaturated fatty acids (LC-PUFAs) including arachidonic acid (AA), due to its role in oxidative stress [36,37]. Insects have high levels of reactive oxygen species (ROS), such as the superoxide anion ($O_2^-$), due to their significant production of ATP in the mitochondrial electron transport chain. These ROS can escape the mitochondria and often react with AA and other LC-PUFAs causing lipid peroxidation leading to damaged cell membranes and possible adduct formation to proteins and nucleic acids, giving rise to additional cell damage [36–38]. In mammals, free AA is released when phospholipase $A_2$ ($PLA_2$) cleaves phospholipids directly from the membrane. In insects

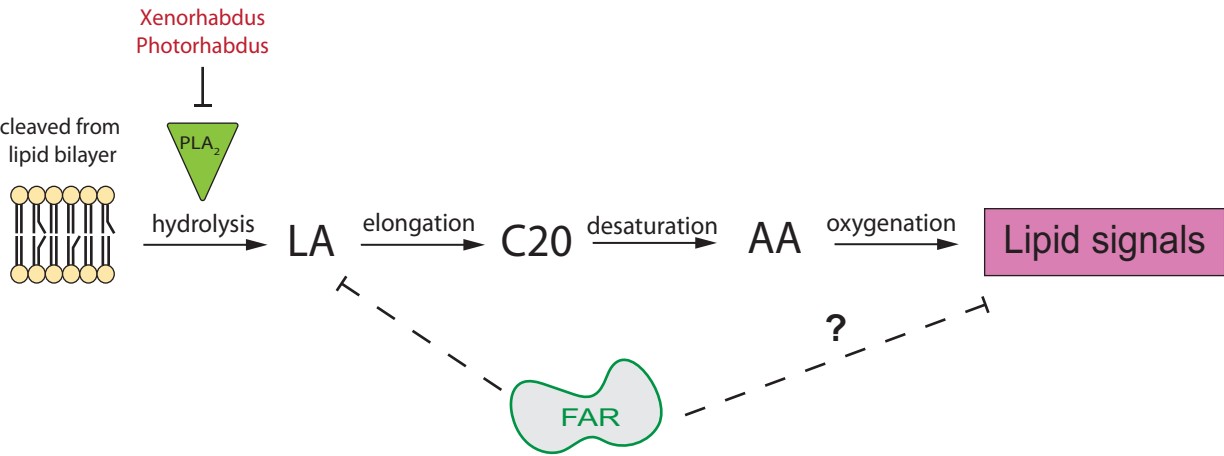

**Fig 6. Key components of lipid biosynthesis in insects.** Our hypothesis is that FAR is sequestering essential fatty acids and/or their upstream lipids to disrupt lipid signaling that is necessary for certain host immune functions. We have shown FARs' ability to bind to L*A in vit*ro which could temporarily disrupt downstream eicosanoid signaling. In insects, phospholipase A2 cleaves linoleic acid (LA) from the lipid bilayer instead of arachidonic acid in mammals. Free LA is then extended via an elongase to a C20 fatty acid. Desaturase oxidizes the C20 to arachidonic acid which can then be oxygenated to yield eicosanoid-like molecules. Eicosanoids are involved in many important functions including gene expression, immune regulation and reproduction. Adapted [36].

however, PLA$_2$ has been predicted to yield free linoleic acid (LA) which is then elongated to AA and subsequently converted to downstream lipids such as prostaglandins and leukotrienes that are essential signaling molecules in the immune response (Fig 6) [36,39]. Our untargeted metabolomics study revealed that phosphatidylcholine (PC) and phosphatidylethanolamine (PE) were reduced in FAR expressing flies (S12 Fig). These are essential phospholipids usually incorporated into the cell membrane that give rise to many downstream lipids that could have diverse functions including immune signaling and regulation [40]. We hypothesize that they are depleted by a compensatory mechanism where more phospholipid is cleaved from the membrane and converted to downstream fatty acids that are likely sequestered and depleted more readily by FARs. Interestingly, our metabolite study of hemolymph shows a low but measurable concentration of AA, with much higher concentrations of C:18 fatty acids including oleic and linoleic acids. These data support the hypothesis that in insects, LA is cleaved directly from the lipid bilayer, providing evidence for the linoleic-to-arachidonic model of C20 biosynthesis in insects. Our data also show that FAR binds tightly to LA and not AA, *in vitro*. We hypothesize that although LA is depleted by FARs, the depletion of downstream lipid products such as 9-HODE and 13-HODE have a more significant effect on immune signaling. In mammals, pro- and anti- inflammatory lipids are produced from identical precursors with the enzymes from each branch competing for substrate [41,42]. Therefore, the regulation of enzyme availability decides the outcome of the immune response. Interestingly, PLA$_2$ is turned on by the Toll and Imd pathways in *D. melanogaster* and inhibited by the bacterial symbionts of EPNs, *Xenorhabdus* and *Photorhabdus* (Fig 6) [27,40,43,44]. It is possible that in *Drosophila* a similar network for substrate and enzyme availability takes place, however the identity of pro- and anti-immune lipid signaling molecules is yet to be determined in this model.

Lipid signals have important roles in mammalian immunity and have been hypothesized to play similar roles in insect immune defenses [45]. We found that *S. carpocapsae* FARs bind to LA and oxidized metabolites of LA *in vitro*, which may suggest their preferred *in vivo* binding partners. Our metabolomics studies showed that FARs deplete linoleic acid, oleic acid, 9(10)- and 12(13)-EpOME, 9,(10)- and 12(13)- DiHOME, and 9- and 13-HODE from the blood of

flies. These lipids are known to modulate diverse physiological functions in mammals and have been shown to regulate immunity in insects [46]. 13-HODE has anti-inflammatory functions in mammalian immunity and is often increased under oxidative stress triggered during a disease response, are depleted in the hemolymph of FAR expressing flies [47,48]. During a mammalian immune response, 9(10)- and 12(13)-EpOME are activated by and interact with inflammatory leukocytes or neutrophils [47]. These data support the hypothesis that FARs modulate host immunity by binding to the oxidation products of LA or its upstream precursors to dampen the immune response (Fig 6). While little is known about how nematodes use FARs and other ES molecules to interact with host immunity, model systems continue to be invaluable in elucidating functional activity and yielding hypotheses that can be further tested.

In summary, this study shows experimentally that FAR-1 and FAR-2 from *S. carpocapsae* dampen the host immune response and provides evidence that nematodes likely utilize other ESPs in conjunction with FARs to modulate host immunity. These data lead us to hypothesize that FARs act by disrupting lipid signaling necessary for immune responses. Deepening our understanding of how nematode parasites evade or suppress host defenses is key to further our development of treatment options for these infections, and may lead to the development of novel treatments for autoimmune disorders.

## Methods

### Fly stock/maintenance

All fly strains were grown on D2 glucose medium from Archon Scientific (Durham, North Carolina) and kept at 25˚C with 50% humidity on a 12h light 12h dark cycle.

### Nematode Infection assays

Infection of OregonR and transgenic *D. melanogaster* adult flies with the infective juveniles of *S. carpocapsae was* performed using the protocol published by Patrnogic *et al*., 2019. Briefly, 2–4-week-old IJs of *S. carpocapsae* collected from infected waxworms using White traps (White, 1927), were used for the infection. Nematode infections were carried out at two densities, 100 worms/fly and 50 worms/fly. Nematodes were suspended in sterile distilled water (250 μl) and added to vials containing four layers of filter papers (Whatman Grade 1, 20 mm). 250 μl of 1% sucrose was introduced to vials as the source of nutrients for the flies. 250 μl of water plus 250 μl of 1% sucrose was used in uninfected controls. Ten anesthetized flies (five males and females) were added to each vial. The vial flug was pushed down to restrict the movement and increase the probability of infection. Vials were incubated at 25˚C. Infection was set up in 6 vials for each nematode density and each fly type. Flies from 3 vials were dissected 24 h post-infection to quantify the nematode infection. Flies from 3 other vials were transferred on food 24 h post-exposure to IJs and survival was monitored daily for 6 days. Dead flies from each day were dissected to see determine successful nematode infection. On the 6th day, all the flies were dissected to check for nematode infection. Infection experiments were replicated three times.

### Plasmid design and assembly

To assemble plasmid OA-1010, the base vector used for generating FAR protein-expressing plasmids, several components were cloned into the piggyBac plasmid OA959C (Addgene #104968) using Gibson assembly/EA cloning. Plasmid OA959C was digested with AvrII and NotI, and the following components were cloned in with EA cloning: an*attP* sequence amplified from plasmid M{3xP3-RFP attP} with primers 1010.C1 and 1010.C2, a 10xUAS promoter

fragment amplified with primers 1010.C3 and 1010.C4 from Addgene plasmid 78897, a p10 3'UTR fragment amplified from Addgene plasmid #100580 with primers 1010.C5 and 1010. C6, and opie2-dsRed marker fragment amplified from Addgene plasmid #100580 using primers 1010.C7 and 1010.C8. The resulting plasmid was then digested with XhoI and PacI, and the coding sequences (CDSs) of various FAR proteins that were codon-optimized for *D. melanogaster* expression and synthesized (GenScript, Piscataway, NJ) were separately cloned in to generate the six final FAR protein-expressing vectors. Specifically, to generate vector OA-1010Av2 (Addgene #176082), the codon optimized CDS of *S. carpocapsae* gene L596_g9608 was amplified with primers 1010.C9 and 1010.C10 from a gene-synthesized plasmid and cloned into the above digested vector using EA cloning. Then, to generate vector OA-1010A (Addgene #176078), OA-1010Av2 was digested with PacI and a G(4)S linker followed by a 30-amino-acid human influenza hemagglutinin (HA) epitope tag was amplified with primers 1010.C11 and 1010.C12 from the *ninaE*[SBP-His] vector and cloned into the above digested vector using EA cloning. To generate vector OA-1010Bv2 (Addgene #176083), the codon optimized CDS of *S. carpocapsae* gene L596_g25050 were amplified with primers 1010.C13 and 1010.C14 from a gene-synthesized plasmid and cloned into the digested OA-1010 vector using EA cloning. Then, to generate vector OA-1010B (Addgene #176079), OA-1010Bv2 was digested with PacI and a G(4)S linker followed by a 30-amino-acid human influenza HA epitope tag was amplified with primers 1010.C15 and 1010.C12 from the *ninaE*[SBP-His] vector and cloned into the above digested vector using EA cloning. To generate vector OA-1010C (Addgene #176080), a fragment containing the codon optimized CDS of *C. elegans* gene *far-8* followed by a G(4)S linker and a 30-amino-acid human influenza HA epitope tag was amplified with primers 1010.C16 and 1010.C12 from a gene-synthesized plasmid and cloned into the XhoI/PacI-digested OA-1010 vector using EA cloning. To generate vector OA-1010D (Addgene #176081), a fragment containing the codon optimized CDS of *A. ceylanicum* gene *Ac-far-1* (maker-ANCCEYDFT_Contig87-pred_gff_fgenesh-gene-3.1) followed by a G(4)S linker and a 30-amino-acid human influenza HA epitope tag was amplified with primers 1010. C17 and 1010.C12 from a gene-synthesized plasmid and cloned into the XhoI/PacI-digested OA-1010 vector using EA cloning. All codon optimization was done by GenScript (Piscataway, NJ) using OptimumGene algorithms.

## Fly transgenesis

Transgenic flies were developed using codon optimized Sc-FAR-1 or Sc-FAR-2 inserted via the PhiC31 site-specific serine integrase method. The transgenic UAS lines were then crossed with several Gal4 drivers from the Bloomington *Drosophila* Stock Center (BDSC), including TubP-Gal4 (#5138; strong, ubiquitous somatic expression), CG-Gal4 (#7011; expressed in the fat body, hemocytes and lymph gland) and He-Gal4 (#8699; expressed in hemocytes). Fly husbandry and crosses were performed under standard conditions at 25˚C. Rainbow Transgenics (Camarillo, CA) carried out all of the fly injections. All constructs were inserted into *attP* line 86Fa (BDSC #24486: y[1] M{vas-int.Dm}ZH-2A w[*]; M{3xP3-RFP.attP'}ZH-86Fa).

## Western blot

A modified version of the abcam general western blot protocol was used. 30 adult HA-tagged transgenic flies were ground up in 200 μL lysis buffer and centrifuged at 20,000g for 20 minutes at 4˚C. The supernatant protein concentration was normalized to 1–2 mg/mL. 20 μL anti-HA magnetic beads (Thermo Scientific 88836) were incubated with 150 μL of supernatant protein sample while shaking for 30 minutes. The sample was then removed, and the beads were washed twice with 300 μL of TBST. The beads were then resuspended in 30 μL Tris-Cl pH 8.0

and 10 μL loading buffer. The samples were heated for 10 minutes at 100°C and electrophoresed on the SDS-PAGE gel. The proteins were transferred onto the immobilon P$^{SQ}$ (Millipore) for 1.5 hours at 50 Volts. The membrane was washed with PBS and blocked with 1% BSA for 30 minutes with shaking. The membrane was washed twice with PBST for 10 minutes and then incubated with HA-tag primary anti-rabbit antibody (Abcam ab236623) for 2 hours with shaking. After incubation, the membrane was washed twice again with PBST for 10 minutes. Then anti-rabbit anti-goat antibody (Abcam ab6721) was added and incubated for 2 hours with shaking. Again, the membrane was washed twice with PBST and then developed with Metal enhanced DAB substrate kit (Thermo Scientific 34065).

## Bacterial stock maintenance

*Streptococcus pneumoniae* was grown by shaking in glass vials with 5 mL tryptic soy (TS) broth (Difco TS broth, catalase, streptomycin) at 37°C with 5% $CO_2$ overnight. The overgrown culture was diluted in catalase (100 μL) and TS to yield a final volume of 20 mL in a flask and incubated shaking until the $OD_{600}$ ~ 0.4 (about 1 hour). The culture was then diluted again to a final volume of 50 mL, with 150 μL catalase, and incubated until the $OD_{600}$ ~ 0.2–0.4 (above 0.5 is no longer in log phase). 5% glycerol was added to the final culture and stored then in 1mL aliquots at -80°C. To use the aliquots, one tube was thawed, spun down at 14,000 rpm for 5 minutes, the supernatant was removed, and the pellet was resuspended in the desired amount of PBS (50–60 μL yields ~ 100,000 CFUs) and serially diluted to yield the appropriate CFU doses. For quantification of CFUs, *S.p.* was plated on TSA agar plates supplemented with 50 mL/L sheep's blood. *Listeria monocytogenes* (serotype 4b, 19115, (ATCC, VA)) was also grown in batches in brain heart infusion (BHI) medium at 37°C in aerobic condition. Cultures were grown overnight in a flask inoculated with a fresh colony and rediluted under log phase (below $OD_{600}$ ~ 0.2) and grown up to the desired $OD_{600}$ (~0.4). The entire volume was transferred to a 50mL centrifuge tube for vortexing. Before freezing, a 5% glycerol solution was added to the culture and 1mL aliquots were stored at -80°C. To use the aliquots, one tube was thawed, spun down at 14,000 rpm for 5 minutes, the supernatant was removed, and the pellet was resuspended in the desired amount of PBS (90–100 μL yields ~ 100,000 CFUs) and serially diluted to yield the appropriate CFU doses. For quantification of CFUs, *L.m.* was plated on BHI plates. *Xenorhabdus innexi* (HGB2121 *att*Tn*7*/Tn*7*-GFP (from pURR25)) was incubated at 27°C shaking in Luria Bertani (LB) broth supplemented with 0.1% sodium pyruvate (sp). Overnight cultures on *X. innexi* were subcultured below log phase ($OD_{600}$< 0.4) and grown back to log phase ($OD_{600}$ 0.4–0.8) and diluted to the desired concentrations before use, as well as streaked on LB+sp plates bi-weekly for storage. LB+sp media was supplemented with 20% glycerol for long term storage of log phase *X. innexi* in -80°C.

## Generation of recombinant proteins

The sequence corresponding to *Sc*FARS-1 *and Sc*FARS-2 was obtained from GenBank [49]. The sequences encoding the mature protein was codon-optimized, synthesized by GenScript and cloned into a modified pET28 vector incorporating an N-terminal hexahistidine tag and a TEV cleavage site. *Sc*FARS-1 *and Sc*FARS-2 were produced recombinantly in *E. coli* BL21-CodonPlus cells (Stratagene) grown in autoinduction medium (Invitrogen) from a 1% inoculum. Following four hours of growth at 37°C and 16 hours at 30°C, the cells were harvested, and the pellet resuspended in 20 mM Hepes pH 8.3, 1 M NaCl, 30 mM imidazole. Cells in suspension were lysed using a French press, insoluble material was removed by centrifugation, and the soluble fraction was applied to a HisTRAP FF 5 mL column (GE Healthcare). Bound *Sc*FARs1 and 2 were eluted with an increasing concentration of imidazole, the His tag was subsequently

removed by TEV cleavage, and they were further purified by gel filtration chromatography on a Superdex 200 16/60 HiLoad column in HBS (20 mM Hepes pH 7.5, 150 mM NaCl). The purity of was assessed at each stage by SDS-PAGE.

## Fly injections, survival and CFUs

For injections and immune assays, 5-7-day-old male flies were anesthetized with $CO_2$ and injected with various CFU doses yielding a total volume of 50 nL precisely using a MINJ-FLY high-speed pneumatic injector (Tritech Research, CA) and individually pulled calibrated glass needles. Flies were injected into the abdomen close to where the thorax meets and slightly ventral from the dorsal-ventral cuticle axis, easily visible below the haltere. Survival studies were carried out for all of the pathogens we tested. After injection of the CFU dose or phosphate buffered saline (PBS) control, flies were placed in vials in groups of 30 with a total of 60 flies per experimental or control group. Flies injected with the human pathogens (*S.p.* and *L.m.)* were kept at 28˚C with 50% humidity compared to flies injected with the insect pathogen *X.i.* which was kept at 25˚C with 50% humidity. The number of dead flies was counted daily and Kaplan-Meier survival curves were generated with GraphPad Prism software with statistics shown as log-rank analysis (Mantel-Cox). Survival experiments were at least triplicated. CFUs were determined by homogenizing a single infected, or buffer-injected fly in 200 μL of PBS, serially diluted and plated on the appropriate agar plates and incubated overnight. Colonies were counted the next day. At least five flies per condition were homogenized for CFU quantification each time an injection experiment was done to measure time 0 CFUs which are representative of all fly strains. All fly strains were injected at the same time for each experimental replicate. Using GraphPad Prism software, results are shown as scatter plots with statistical significance analyzed using an unpaired t-test.

## PO & disseminated melanization

Flies were injected with 1,000 CFUs of *L. monocytogenes* to elicit an immune induced melanization cascade. Phenoloxidase activity was measured as previously described [50, 51]. To collect hemolymph, 20–30 flies 4 hours post injection (p.i.) were pricked through the thorax and placed in a pierced 0.5 μL Eppendorf tube and covered with glass beads, then placed inside a 1.5 μL Eppendorf tube containing 30 μL of 1x protease inhibitor cocktail (Fisher, PI78429). Samples were centrifuged at 10,000 rpm for 20 minutes at 4˚C. Protein concentrations were measured with Bradford assay (Bio-rad, 5000006) and then diluted in phosphate buffered saline (PBS) to a concentration of 15 μg/uL and a total volume of 100 μL. Using a clear 96-well plate, each well contained 160 μL L-Dopa (3 mg/mL) dissolved in phosphate buffer (37.5% 1 M potassium phosphate, 62.5% 1 M sodium phosphate, pH 6.5), 35 μL of hemolymph sample and 5 μL $CaCl_2$ (20 mM). PO activity was measured by kinetic reads at 29˚C at 492 nm every minute for 60 min with 5 seconds of shaking between reads. The OD of a blank control was subtracted from all biological values. Experiments were replicated five times with three technical replicates per experiment. Data were plotted as mean+SEM by taking the peak OD value (timepoint ~ 30 min). Statistics shown as an unpaired t-test was done in GraphPad Prism. For disseminated melanization, flies were observed for melanin deposits in the posterior and anterior abdomen, thorax, head and eyes four days p.i. with *L.m.* An individual was considered to show disseminated melanization if it had two or more deposits of melanin, one often at the wounding site and another either underneath the cuticle or in deeper tissues as previously described [28]. Data were graphed as percent of the population infected that was melanized by day four p.i. as the mean+SEM. Experiments were replicated three times with 40 individuals

per experimental condition per experiment. Statistics shown as unpaired t-test was done in GraphPad Prism.

## Antimicrobial peptide gene expression—qPCR

Total RNA was extracted from 15 *S. pneumonia* infected flies per strain 24 hours post-injection using Trizol reagent (Molecular Research Center, Inc; Cincinnati, Ohio) according to the manufacturer instructions. Integrity of RNA was confirmed by observing bands on an agarose gel and concentration was determined by nanodrop. Reverse transcription of RNA using ProtoScript II First Strand cDNA synthesis kit (New England BioLabs, NE, E6560L) following the manufacturer protocol, in a MultiGene OptiMax Thermal Cycler (Labnet international, NJ). The qRT-PCR was done with a CFX Connect Bio-Rad system with Perfecta SYBR green super-mix (QuantaBio, MA) and gene specific primers for *Defensin*, *Drosomycin*, and *Tubulin* (Integrated DNA Technologies, IA). Experiments were carried out with three technical replicates and repeated four times with plots shown as bar graphs with individual points representing each replicate. Statistics shown as One-way ANOVA done in GraphPad Prism.

## In vitro binding -

The fatty acid- and retinol- binding preferences of Sc-FAR *in vitro* was measured by utilizing the saturated fatty acid fluorescent probe 11-(Dansylamino) undecanoic acid (DAUDA) (Sigma-Aldrich, USA) and retinol as previously described [9,31]. Fluorescent emission spectra for Sc-FAR bound to DAUDA (1 µM) and retinol (40 µM) were measured at 25°C in a black-walled 96-well plate yielding a total volume of 200 µL with an excitation wavelength of 345 nm and 350 nm respectively. When DAUDA is encompassed by a binding protein a 50nm blue shift is observed with an excitation wavelength of 345nm. The equilibrium dissociation constant (Kd) for Sc-FAR bound to DAUDA or retinol was estimated by adding increasing concentrations of Sc-FAR, in 1 or 2 µM increments, to 1 µM DAUDA in PBS and 40 µM retinol in PBS. The data were normalized to the peak fluorescence intensity of DAUDA or retinol bound to FAR (yielding a value of 1) and corrected for background fluorescence of PBS alone for each concentration. The data were then plotted as relative fluorescence and a nonlinear fit via the one site-specific system was used to find the Kd value in GraphPad Prism. Competition studies were done by measuring the decrease in peak emission of DAUDA in the presence of another fatty acid in 10-fold excess. Oleic, linoleic and arachidonic acid were tested along with 9,(10)- and 12,(13)- EpOME and 13-HODE(Cayman Chemicals, MI). All competition experiments were replicated 3 times with 3 technical replicates per experiment, plotted as bar graphs with mean+SEM and individual points for each replicate, analyzed by an unpaired t-test. All fatty acids and DAUDA were stored in -20°C and freshly diluted before each experiment starting with a working solution of either 10 or 100 µM in 100% ethanol and then subsequently diluted in PBS to achieve the appropriate working solution.

## Metabolomics

**Whole fly untargeted -.** Flies were transferred to a 2 mL bead mill tube and weighed, the range was 131 mg to 302 mg. Ice-cold extraction solvent (20:20:30:30 water:IPA:ACN:MeOH) was added, 3.32 µL/1 mg sample, and samples were homogenized in a liquid nitrogen cooled bead mill, 6, 10 s cycles at 5 m/s. After centrifugation for 15 min at 16,000 x g at 4 C, the supernatant was analyzed by LC-MS. LC-MS metabolomics analysis was performed at the UC Riverside Metabolomics Core Facility as described previously [52]. Briefly, analysis was performed on a Synapt G2-Si quadrupole time-of-flight mass spectrometer (Waters) coupled to an I-class UPLC system (Waters). Separations were carried out on a CSH phenyl-hexyl column (2.1 x

100 mm, 1.7 μM) (Waters). The mobile phases were (A) water with 0.1% formic acid and (B) acetonitrile with 0.1% formic acid. The flow rate was 250 μL/min and the column was held at 40˚ C. The injection volume was 2 μL in positive ion mode and 4 μL in negative ion mode. The gradient was as follows: 0 min, 1% B; 1 min, 1% B; 8 min, 40% B; 24 min, 100% B; 26.5 min, 100% B; 27 min, 1% B. The MS was operated in positive ion mode (50 to 1200 m/z) with a 100 ms scan time. MS/MS was acquired in data dependent fashion. Source and desolvation temperatures were 150˚C and 600˚C, respectively. Desolvation gas was set to 1100 L/hr and cone gas to 150 L/hr. All gases were nitrogen except the collision gas, which was argon. Capillary voltage was 1 kV in positive ion mode and 2 kV in negative ion mode. A quality control sample, generated by pooling equal aliquots of each sample, was analyzed every 4–6 injections to monitor system stability and performance. Samples were analyzed in random order. Leucine enkephalin was infused and used for mass correction. Untargeted data processing (peak picking, alignment, deconvolution, integration, normalization, and spectral matching) was performed in Progenesis Qi software (Nonlinear Dynamics). Data were normalized to total ion abundance. To aid in the identification of features that belong to the same metabolite, features were assigned a cluster ID using RAMClust [53]. An extension of the metabolomics standard initiative guidelines was used to assign annotation level confidence [54,55]. Annotation level 1 indicates a match to an in-house library. Level 2a indicates an MS and MS/MS match to an external database. Level 2b indicates an MS and MS/MS match to the Lipiblast in-silico database [56] or an MS match and diagnostic evidence, such as the dominant presence of an m/z 85 fragment ion for acylcarnitines. Level 3 indicates an MS and partial MS/MS match to an external or in-house database. Several mass spectral metabolite databases were searched against including Metlin, Mass Bank of North America, and an in-house database. Statistical analyses were performed, and figures generated using R.

**Hemolymph only (WashU) -.** Hemolymph from 200 5-7-day old male flies was extracted as previously described in [57]. Briefly, flies were pierced through the thorax with a tungsten needle. Flies were placed in a pierced 0.5 mL Eppendorf tube within a 1.5 mL Eppendorf tube containing 20μL of 10x protease inhibitor cocktail and centrifuged for two rounds of 10 minutes at 10,000 rpm at 4˚C with a gentle mixing in between rounds. The supernatant of the collected hemolymph was centrifuged for 10 minutes at 14,000 rpm to remove cells and debris. The supernatant was flash-frozen in liquid nitrogen and stored at -80˚C until prepped for metabolomics experiments.

Each of the pooled drosophila sample was initially homogenized with 260 μL of methanol and 30 μL of water containing 2ng each of deuterated oxidative metabolites (9, 10-EpOME-$d_4$, 12.13-EpOME-$d_4$, 9, 10-DiHOME-$d_4$, 12, 13-DiHOME-$d_4$, 9-HODE-$d_4$, and 13-HODE-$d_4$) as well as deuterated fatty acids (100ng; AA-$d_8$ and 1000ng; linoleic acid (LA)-$d_4$) as the internal standards. Fatty acids as well as the internal standards were derivatized from 40 μL of the homogenate with 50 mM DMAPA, 100mM EDC, and 100 mM DMAP. Four- to five-point calibration standards of oxidative metabolites and fatty acids containing their deuterated internal standards were also prepared for the absolute quantification.

The sample analysis for the oxidative metabolites was performed with a Shimadzu 20AD HPLC system coupled to a tandem mass spectrometer (API-6500⁺Qtrap: Applied Biosystems) operated in MRM positive ion mode. For fatty acid analysis another Shimadzu 20AD HPLC system coupled to the API-4000Qtrap mass spectrometer were used in MRM positive ion mode. All samples were injected in duplicate for data averaging. Data processing was conducted with Analyst 1.6.3 (Applied Biosystems). Metabolomic analysis of fly hemolymph was performed by the Metabolomics Facility of Washington University (St. Louis, MO).

**Statistics.**   All statistics were done with GraphPad Prism 8.4. Statistical significance indicated with asterisks indicating the following p-value cut offs: 0.05–0.033*, 0.033–0.002**, 0.002–0.0002*** and <0.0001****.

## Supporting information

**S1 Fig. *D. melanogaster* has little to no survivability to *S. carpocapsae* infections.** Flies were exposed to a dose of 100 IJs/fly, using *S.c.* IJs. The survival rate equals the infection rate. Once a fly is infected with even just one IJ it cannot survive the infection. Percent mortality overlaps with percent infected. As with bacterial infections, various fly strains interact with infections differently. Each line represents at least 90 flies.
(EPS)

**S2 Fig. Dose response of various pathogens in wild-type OregonR flies was used to assess virulence patterns and to identify the LD$_{30}$, (dose that leads to 30% death of the population within the first 1–5 days depending on the pathogen) an optimal dose to measure variations in immunity.** 5-7-day-old male flies were used for all injections with phosphate buffered saline (PBS) shown as the vehicle control. A) 100 to 100,000 CFUs of *Streptococcus pneumoniae* were injected, LD$_{30}$ identified as 7,000 cells. B) 10 to 100,000 cells of *Listeria monocytogenes* were injected, LD$_{30}$ identified as 1,000 cells. C) 100 to 100,000 cells of the insect pathogen *Xenorhabdus innexi* were injected, 25,000 and 50,000 cell doses were chosen for study. All raw data available in supplemental materials.
(EPS)

**S3 Fig. We performed additive recombinant FAR-1 and FAR-2 injection experiments with the LD$_{30}$ of *S. pneumoniae* and found no difference between a 250 ng dose of FAR-1 or FAR-2 and a 125 ng FAR-1 plus 125 ng FAR-2 combined dose showing that FAR does not have an additive effect.** The graph below shows at least 180 flies per line with statistical significance shown as a log-rank test.
(EPS)

**S4 Fig. Recombinant FAR elicits a specific effect on the outcome of a bacterial infection.** As a control to validate the effects of recombinant FAR proteins on immunity, denatured *S. carpocasae* FARs were co-injected with 7,000 CFUs of *S. pneumoniae*. A) Denatured Sc-FAR-1 (250ng) co-injected with *S.p.* shows no significant difference from *S.p.*-only injected flies. B) Denatured Sc-FAR-2 (250ng) co-injected with *S.p.* shows no significant difference from *S.p.*-only injected flies. Statistics shown as Log-rank test. All raw data available in supplemental materials. Experiments not found to be significantly different from bacteria-only controls were marked ns.
(EPS)

**S5 Fig. Overview of the UAS-Gal4 genetic crosses of FAR expressing and control flies.** The sequence of FAR-1 or -2 was inserted into the 86Fa fly strain which displays a fluorescent red body and eye color and crossed with various Gal4 promoter strains. As a control, the 86Fa fly was crossed with the Gal4 driver flies as well as the FAR-1 or -2 transgenic flies were crossed with a *w*$^{1118}$ strain. The male flies yielding the appropriate genotypes were tested for their immunodeficiencies after inducting an immune response with a bacterial injection into the abdomen.
(EPS)

**S6 Fig. Western blot shows *in vivo* production of FAR proteins.** FAR is expressed when driven with the Gal4 driver in the first 4 wells (rows 1–4). Control flies not expressed with

UAS-Gal4 expression system do not show the presence of FAR proteins (rows 5–8).
(EPS)

**S7 Fig. FAR proteins from other species of nematodes show similar negative effects on the outcome of a _S. pneumoniae_ infection.** Flies expressing FAR-8 from the free-living nematode _Caenorhabditis elegans_ or a FAR from the mammalian hookworm _Ancylostoma ceylanicum_ (maker-ANCCEYDFT_Contig87-pred_gff_fgenesh-gene-3.1) with the CG driver had a significant decrease in survival after injected with 7,000 cells of _S.p_. Statistics shown as Log-rank test. All raw data available in supplemental materials.
(EPS)

**S8 Fig. Lifespan of FAR transgenic flies expressed with the fat-body and hemocytes specific driver CG.** Lifespan is not altered within the timeframe where immunity studies took place (day 0 to 20). Statistics shown as Log-rank test. All raw data available in supplemental materials.
(EPS)

**S9 Fig. Genetic control of 86Fa transgenic strains.** As a control to validate the specific effects on immunity after promoting the FAR transgenics, FAR-1 & -2 expressing flies were crossed with $w^{1118}$ flies. There was no significant difference observed between all genotypes validating the specific effects of FAR. All raw data available in supplemental materials.
(EPS)

**S10 Fig. When mCherry is overexpressed by the CG-Gal4 driver, there is no significant reduction in survival as compared to control crosses.** All lines represent at least 180 flies.
(EPS)

**S11 Fig. Simplified overview of Drosophila immunity.** Detection of pathogens elicits an array of interconnected innate immune responses specifically divided into the humoral, or systemic, and the cellular response. Humoral immunity leads to the production of antimicrobial peptides (AMPs) downstream of either the Toll or Imd pathways. Cellular immunity is carried out by different types of hemocytes that surround and kill invading microbes. Adapted from [24,58].
(EPS)

**S12 Fig. Volcano plot comparing relative abundance of metabolites in _CG_>_FAR-1_ expressing and _CG_>+ control flies.** Phosphatidylcholines (PC), phosphatidylethanolamines (PE) and multiple fatty acids were found to be more abundant in control flies, meaning they are depleted in FAR expressing flies as well as potential binding partners, or upstream compounds of FARs binding partners. All raw data available in supplemental materials. LC-MS analysis depicted in this figure was performed at the UC Riverside Metabolomics Core Facility.
(EPS)

**S13 Fig. _In vitro_ binding properties of _S. carpocapsae_ FARs to fatty acids.** A&B) Plots of increasing concentration of FAR binding to 1μM 11-(Dansylamino) undecanoic acid (DAUDA) in PBS reveal a $K_d$ in the 10 μM range. $K_d$ was estimated by using the best-fit in Graphpad Prism. C&D) When DAUDA is bound to FAR (13 μM) _in vitro_, a ~50nm blue shift is observed in the peak. E&F) Full curve of _in vitro_ competition assays shows a decrease in fluorescence when linoleic acid (10 μM) binds to and therefore displaces DAUDA (1 μM) from the fatty-acid binding pocket of FAR. All raw data available in supplemental materials.
(EPS)

**S14 Fig.** *In vitro* **binding properties of** *S. carpocapsae* **FARs to retinol.** A&B) Plots of increasing concentration of Sc-FAR binding to 40μM retinol in PBS reveal a $K_d$ in the 1 μM range. $K_d$ was estimated by using the best-fit in Graphpad Prism. C&D) When retinol is bound to FAR (5 μM) *in vitro*, the peak fluorescence is greatly increased. All raw data available in supplemental materials.
(EPS)

**S15 Fig. FARs' role in increased mortality and microbe load is not limited to the $LD_{30}$ dose.** FAR-1 and -2 were expressed with the CG driver and injected with 100 to 100,000 CFUs of *S.p.* A&C) Survival was measured daily showing that FARs significantly increase mortality rate in most doses. B&D) In the presence of FAR, various doses of *S.p.* also cause an increase in microbe growth 24 hours p.i. Dotted lines show *CG>+* and solid lines show *CG>FAR-1* or *-2*. Statistics shown as Log-rank tests for survival curves and unpaired t-tests for microbe growth. All raw data available in supplemental materials.
(EPS)

**S1 Data. Excel file contains all the raw data used in this research study.** The data used for each figure is included on a separate tab, organized by figure number.
(XLSX)

## Acknowledgments

We thank Joshua Mallilay, Nikhil Prabhakar, and Priscila Robles, for their assistance with experimentation, fly husbandry and maintenance, and Bloomington *Drosophila* Stock Center (NIH P40 OD018537) for providing fly stocks. We thank Jay Kirkwood and the UC Riverside Metabolomics Core Facility, and the Metabolomics Facility at Washington University for their assistance with parts of this study.

## Author Contributions

**Conceptualization:** Sophia C. Parks, Susan Nguyen, Shyon Nasrolahi, Omar S. Akbari, Adler R. Dillman.

**Data curation:** Sophia C. Parks, Adler R. Dillman.

**Formal analysis:** Sophia C. Parks, Shyon Nasrolahi, Chaitra Bhat, Damian Juncaj, Dihong Lu, Harpal Dhillon, Hideji Fujiwara, Adler R. Dillman.

**Funding acquisition:** Adler R. Dillman.

**Investigation:** Sophia C. Parks, Susan Nguyen, Shyon Nasrolahi, Chaitra Bhat, Damian Juncaj, Dihong Lu, Raghavendran Ramaswamy, Harpal Dhillon, Anna Buchman.

**Methodology:** Sophia C. Parks, Anna Buchman, Omar S. Akbari, Naoki Yamanaka, Martin J. Boulanger, Adler R. Dillman.

**Project administration:** Adler R. Dillman.

**Resources:** Raghavendran Ramaswamy, Harpal Dhillon, Hideji Fujiwara, Omar S. Akbari, Naoki Yamanaka, Martin J. Boulanger.

**Supervision:** Naoki Yamanaka, Martin J. Boulanger, Adler R. Dillman.

**Writing – original draft:** Sophia C. Parks, Naoki Yamanaka, Martin J. Boulanger, Adler R. Dillman.

**Writing – review & editing:** Sophia C. Parks, Anna Buchman, Omar S. Akbari, Naoki Yama-naka, Martin J. Boulanger, Adler R. Dillman.

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
