## [Decision Letter · Decision Letter 0]

18 May 2021

Dear Dr. Dillman,

Thank you very much for submitting your manuscript "Parasitic nematode fatty acid- and retinol-binding proteins compromise host immunity by interfering with host lipid signaling pathways" for consideration at PLOS Pathogens. As with all papers reviewed by the journal, your manuscript was reviewed by members of the editorial board and by several independent reviewers. In light of the reviews (below this email), we would like to invite the resubmission of a significantly-revised version that takes into account the reviewers' comments.

While the reviewers found this work intriguing, there is a concern about the in vivo physiological relevance of the over expressions data. Please address the issues raised by the reviewers in Figures 1 and 4 as well as address the fatty acid modulation induced by natural Sc infection.

We cannot make any decision about publication until we have seen the revised manuscript and your response to the reviewers' comments. Your revised manuscript is also likely to be sent to reviewers for further evaluation.

Sincerely,

Keke C Fairfax, PhD

Guest Editor

PLOS Pathogens

P'ng Loke

Section Editor

PLOS Pathogens

Kasturi Haldar

Editor-in-Chief

PLOS Pathogens

orcid.org/0000-0001-5065-158X

Michael Malim

Editor-in-Chief

PLOS Pathogens

orcid.org/0000-0002-7699-2064

While the reviewers found this work intriguing, there is a concern about the in vivo physiological relevance of the over expressions data. Please address the issues raised by the reviewers in Figures 1 and 4 as well as address the fatty acid modulation induced by natural Sc infection.

Reviewer's Responses to Questions

**Part I - Summary**

Reviewer #1: Overall, the authors have meticulously shown the effect of Sc-FARs in immunomodulating the host immunity during infection and specifically in this paper by modulating the lipid signaling molecules.

Reviewer #2: The authors provide evidence for an immunesuppressive role for nematode derived FARs in vivo using a fly model, by showing that recombinant FAR administration or ectopic expression of FARs in drosophila flies renders them less resistant to bacterial infection. This was associated with a reduction in several immune response readouts. They additionially show that FAR1 and FAR2 can bind several lipid mediators.

The main strength and novelty of this study is the use of an in vivo model to assess the in vivo portential of FARs to modulate host immune responses. However, it was already shown that FARs bind fatty acids, that they can be immunemodulatory and therefore this study, while solidly executed, does bring reltively little new to the able. Moreover, the physiological relevance of the concentartions of FARs reached in vivio following injection or by expression in these flies remains unclear. Finally the relevance of the FAR binding to the FAs for their effect is unclear, but would be important to provide true novel mechanistic insight

Reviewer #3: The current study by Parks et al is an exciting study on Fatty acid and retinol binding proteins in a parasitic nematode species of insects, S. carpocapsae. The central idea is that FARs are widely expressed across multiple species and these proteins are found within E/S products that would impact host biology. Indeed the authors show that either by recombinant protein injection of FAR's or transgenic overexpression in Drosophila, that FARS impair the response to bacterial challenge and lead to reduced host survival, which. is correlated with reduced phenoloxidase activitiy, melanization and AMP activity. They go on to demonstrate binding activity of FARS and show that at least for FAR-1, that over-expression. in teh. fly reduces metabolite abundance in the hemolymph. This is a comprehensive and potentially highly impact. ful study

**Part II – Major Issues: Key Experiments Required for Acceptance**

Reviewer #1: no major issues were identified.

Reviewer #2: 1) the physiological relevance of the findings remains unclear. specificically what is the physiological relevance of 250ng FAR administration? along the same what is concentration is reached of FARs in vivo in genetically modified Dm flies? Is this similar to what is found during natural Sc infection? These questions should be addressed by determining FAR concentrations during a natural Sc infection and in flies genetiically modified to express these FARs.

2) the changes in lipid composisition in vivo following FAR exposure are interesting. To establish whether this of physiological relevance the author should assess whether during a naturla Sc infection a similar change in lipid compostion is observed.

3) The observation that FARs bind certain FAs is clearly shown, but whether this functionally linked to their immunesuppressive effect remains speculative. Therefore the authors should take advantage of their expression systems to generate FAR-variants in which putative FA-binding domains are mutated. This mutant FARs should then be tested for their ability to suppress immune responses and compromise bacterial clearance in the flies.

Reviewer #3: Most of my issues have to do with completeness of the data sets and issues with data display and interpretation.

1. FIg 1B, it seems like 20 and 50ng recombinant FAR-2 enhances survival which is not discussed at all and is contrary to the main hypothesis. Is this data consistent? The authors need to make an interpretation

2. There are non experiments that use a transgenic over-expression of an irrelevant protein instead of just empty vector as control

3. FIg. 4 is a critical part of this work. Why is only FAR1 expressing flies shown. This definitely needs to include FAR2 because there was different efficacy between the two genes

**Part III – Minor Issues: Editorial and Data Presentation Modifications**

Reviewer #1: I wonder if the authors have investigated whether the diminished Melanization, phenoloxidase and antimicrobial peptide activity is due to the effect of the FAR proteins on the Toll, Imd and/or TGF-b pathways upon bacterial infections. It was shown that TGF-β signaling in D. melanogaster is also regulated after parasitic nematode infection (PMID: 31316388). A discussion of these aspects can expand on the mechanism(s) by which the FAR proteins are the most effective, and thus possibly support similar research focused on immunomodulation in other host-parasite systems.

Reviewer #2: 1) Are there no significant differences in figure 1C?

2) they authors should better discuss the similarities or differences in function between FAR1 and 2. Would there be an additive effects of FAR1 and 2 are administered simultaniously?

Reviewer #3: 4. Gene alignment data should be included for the FAR genes across species.

5. Why is L. monocytogenes usedf or the data panels in Fig. 3,?

6. Throughout the manuscript, the labelling of the figure legend is confusing and is difficult to discern, they should be reordered from left to right in ascending concentration

PLOS authors have the option to publish the peer review history of their article (what does this mean?). If published, this will include your full peer review and any attached files.

Reviewer #1: No

Reviewer #2: No

Reviewer #3: No
---

## [Editor Report · Decision Letter 1]

11 Oct 2021

Dear Dr. Dillman,

We are pleased to inform you that your manuscript 'Parasitic nematode fatty acid- and retinol-binding proteins compromise host immunity by interfering with host lipid signaling pathways' has been provisionally accepted for publication in PLOS Pathogens.

Best regards,

Keke C Fairfax, PhD

Guest Editor

PLOS Pathogens

P'ng Loke

Section Editor

PLOS Pathogens

Kasturi Haldar

Editor-in-Chief

PLOS Pathogens

orcid.org/0000-0001-5065-158X

Michael Malim

Editor-in-Chief

PLOS Pathogens

orcid.org/0000-0002-7699-2064

Thank you for you revisions, the manuscript is now significantly stronger and a great contribution to the field.
---

## [Editor Report · Acceptance letter]

22 Oct 2021

Dear Dr. Dillman,

We are delighted to inform you that your manuscript, "Parasitic nematode fatty acid- and retinol-binding proteins compromise host immunity by interfering with host lipid signaling pathways," has been formally accepted for publication in PLOS Pathogens.

Best regards,

Kasturi Haldar

Editor-in-Chief

PLOS Pathogens

orcid.org/0000-0001-5065-158X

Michael Malim

Editor-in-Chief

PLOS Pathogens

orcid.org/0000-0002-7699-2064